# Are Sustainability Indices Infected by the Volatility of Stock Indices? Analysis before and after the COVID-19 Pandemic

**Manuel Carlos Nogueira** [1,2,*] and **Mara Madaleno** [2]

1 ISPGAYA–Higher Polytechnic Institute of Gaya, Avenida dos Descobrimentos 303, Santa Marinha, 4400-103 Vila Nova de Gaia, Portugal

2 GOVCOPP–Research Unit in Governance, Competitiveness and Public Policy, Department of Economics, Management, Industrial Engineering and Tourism (DEGEIT), University of Aveiro, 3810-193 Aveiro, Portugal

\* Correspondence: mnogueira@ispgaya.pt

**Abstract:** Considering the growing importance of sustainable investments worldwide, we explore the volatility transmission effects between the EURO STOXX Sustainability Index and the stock market indexes of its stocks. Using daily index return data, during 2000–2022, covering the COVID-19 pandemic, Multivariate Generalized Auto-Regressive Conditional Heteroskedasticity (MGARCH) models are used to explore if volatility effects of the stock indices felt during the pandemic implied any evolution in the effects already felt between the volatilities existing in these stock indices and the effects of stock market indices' volatility over the sustainability index. Results point to the great dependence that the sustainability index has on stock index movements. The volatility felt in stock indices during the pandemic period did not become decisive in reversing a previous correlation trajectory between the stock market and sustainability indexes. Provided that sustainability is not observed exclusively in financial and economic terms, but in a triple bottom line context (including the social and environmental sides), we should not verify a high influence of stock market indexes over the sustainability index, as the results point out. Policymakers and investors should be aware of the high influence and take measures to turn the sustainability index more independent.

**Keywords:** EURO STOXX sustainability index; European stock indexes; COVID-19 pandemic; MGARCH models; volatility spreads

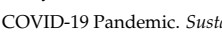



## 1. Introduction

Sustainable living in present times is a necessity for mankind. Investors face dilemmas at the time of choosing the best portfolio composition, able to ensure high returns and low risks [1,2]. Talan and Sharma [3] present a review of 213 articles, concluding that the investment strategy focusing on Environmental, Social, and Governance (ESG) is crucial to ensure sustainable investments. Recently, Umar et al. [4] examined the impacts of the COVID-19 pandemic on the volatility of the ESG leaders' indices through wavelets. The authors call attention to the diversification potential of ESG investments during the pandemic and highlight the potential role of designing cross-geography hedge strategies using data from the USA, Europe, China, and the Emerging Markets' sustainability indexes. Sharma et al. [1] results suggest short-term uni-directional causality from sustainable indexes to conventional indexes and bi-directional causality in the medium and long term. As such, portfolio and fund managers are advised to invest in sustainable indexes to avail of higher returns over a longer period.

There is also growing importance of socially responsible investments in the wake of climate change mitigation goals reinforced by COP26 [5]. Moreover, with the spread of the COVID-19 pandemic, traditional energy stock prices such as oil hit huge values, reinforced by the recent Ukraine–Russia conflict, despite the raised awareness of people about environmental protection and climate change focus [6]. Most of the empirical

research performed in this regard uses wavelets to analyze the impact and spreads among sustainability indexes [1,4,5]. Iqbal et al. [5] conclude for the 14 sustainability indices analyzed between January 2005 and March 2021, through daily returns and volatility, that Germany, France, the Netherlands, and the UK are the primary transmitters, and that negative returns transmit more strongly than positive, being significantly enhanced during the COVID-19. By also resorting to wavelets, Andersson et al. [7] document a significant bidirectional causal relationship between ESG, conventional, and ethical equity portfolio returns.

Using multivariate GARCH models, Zhang et al. [6] study the dynamic connectedness between the ESG, renewable energy, green bond, sustainability, and carbon emission futures stock indexes. The authors conclude that the highest hedging effect is achieved by assuming a long position in carbon futures and a short one in the renewable energy index. Also, Piserà and Chiappini [8] use multivariate GARCH models supporting the superior risk hedging properties of ESG indices over cryptocurrency, but no safe haven properties were attributed to ESG, bitcoin, gold, and WTI (West Texas Intermediate). Thus, sustainable investment performance is still heterogeneous worldwide, although superior risk-adjusted opportunities are identified for investors by including sustainable investment practices in their portfolios, comparing the conditional correlation and volatility behavior of sustainable indices and typical indexes by applying the Dynamic Conditional Correlation–GARCH model offering critical insights to potential investors across developed and developing countries [9,10]. Despite the similarities of this study with ours, the authors use only 5 years (January 2013 to December 2017), not focusing attention on the European market, nor even on the effects of COVID-19. The results of Jain et al. [11] indicate that there is no significant difference in the performance between sustainability indices and the traditional conventional indices, being a good substitute for the latter for the same period and using the same data and methodology as in Sharma et al. [10].

Financial markets have been severely affected worldwide by the global COVID-19 pandemic [4–12]. The pandemic has adversely affected the diversification attributes of various asset classes [2]. This happened mostly due to the much-synchronized exhibition behavior of financial assets through the COVID-19 bear market and during its initial recovery period [4]. As for the effects of COVID-19, Latif et al. [13] use daily data from Canada and the USA to advise investors that rather than investing in stocks they should invest in gold during the period of analysis, due to the increased uncertain negative effects on stock markets. The findings of Li et al. [14] suggest that COVID-19 fear is the ultimate cause driving public attention and stock market volatility, also suggesting investments in the gold market rather than in the stock market. Umar and Gubareva [12] apply wavelet analysis and point to the diversification potential of ESG investments during a pandemic, such as that of COVID-19, and report the potential role of cross-geography hedging strategies. The authors study how the social media coverage of the COVID-19 pandemic influenced the ESG volatility of world indices (World, USA, Europe, China, and Emerging markets' ESG indexes).

Clean energy investments became an attractive investment for investors, fueled by the rising importance of climate-related risks to the global economy and contribution benefits to financial stability and performance [2,15]. Understanding the importance of alternative investments can provide attractive returns and useful hedging strategies [2,4,16]. Even before the pandemic, the academic financial literature started focusing on factors such as sustainability, ethics, environmentally friendly products, and stocks, corporate social responsibility, and financial decisions [17–22]. As a consequence, ESG securities emerged [4].

In the present article, we intend to verify the possibility of the stock indices of European countries that have companies that are part of the EURO STOXX Sustainability Index, contaminating the volatility of this index. For this purpose, we used the daily valuations of these stock indices (those from Portugal, The Netherlands, Italy, Germany, Spain, France, Belgium, Austria, Ireland, Finland, and Luxembourg) and the daily value of the EURO

STOXX Sustainability Index, for the period between 3 July 2000, and 28 February 2020 (i.e., a period greater than 20 years before the COVID-19 pandemic). Additionally, to understand if there was any change in the contribution to the model after the COVID-19 pandemic declaration issued by the World Health Organization, we estimated a second model with all data between 3 July 2000 and 30 September 2022. For this volatility transmission exploration, we resorted to multivariate generalized auto-regressive conditional heteroscedastic models (MGARCH).

Many articles analyze the return and volatility attributes of these assets [23–27]. First, we contribute to this strand of literature by investigating volatility spillovers from traditional stock market indexes of the companies' stocks belonging to the sustainability index, the EURO STOXX Sustainability Index. Secondly, the recent COVID-19 pandemic presented unique challenges and inspired the emergence of literature exploring the impact of this pandemic on financial markets. Our research contributes to the incipient and sparse literature concentrating on volatility spillovers, accounting for the pre- and during-COVID-19 effects using a larger period of analysis, and until more recently (last day of September 2022). Therefore, since our sample covers the recent global crisis caused by the pandemic, our findings can provide valuable insights for socially responsible investors. Third, our article contributes significantly to the emergent literature by having a larger data span and analyzing the volatility interactions of eleven European common stock market indexes to the EURO STOXX Sustainability Index, from which and to which their companies contribute. Finally, we use a multivariate GARCH model, which is one of the most useful tools for analyzing and forecasting the volatility of time series when volatility fluctuates over time. It is this feature that demonstrates its availability in modeling the co-movement of multivariate time series with varying conditional covariance matrices.

The rest of the article develops as follows. Section 2 presents the methodology employed, whereas data, variables, statistics, and correlations are presented in Section 3. In Section 4 model testing and previous analysis are highlighted and discussed, whereas results are to be exposed in Section 5. Finally, Section 6 discusses the results and Section 7 is where conclusions are evidenced.

## 2. Methodology: Multivariate Generalized Auto-Regressive Conditional Heteroskedasticity (GARCH) Models

In this paper, we will use multivariate GARCH models, which in addition to allowing links between stock market indices, also model returns for the EURO STOXX Sustainability Index. These are useful tools to analyze fluctuating volatility over time of time series, allowing one to model their co-movement by implying a varying conditional covariance matrix.

Since the seminal development of the univariate ARCH models by Engle [28] and GARCH models by Bollerslev [29], the scientific literature has used these models to study the volatility in time series. The multivariate GARCH family models are widely used to verify, quantify, and monitor the volatility and dynamics of macroeconomics and financial time series correlations (and for other types of time series) effectively [30]. With a similar opinion, Cavicchioli [31] argues that the models of the GARCH family (The GARCH model family is completed by successive methods such as IGARCH, TGARCH, EGARCH, PARCH, CGARCH, FIGARCH, FIEGARCH, SWARCH, GJR-GARCH, NA-GARCH, AGARCH, MGARCH, among others) are more effective than the classic time series models, or the vector autoregressive moving-average (VARMA) process, in capturing the effects and empirical characteristics of the time series.

To Bonga-Bonga [32] and many other authors, the contagion effect in financial markets has been immensely proven in empirical terms in the literature. However, as far as we know, this is the first time that someone has studied whether the effect and contagion of the financial markets are verified by the EURO STOXX Sustainability Index. The term financial contagion generally refers to transfers of shocks from financial crises [32]. Although there is no consensual definition in the literature on the exact concept of financial contagion, it

is used by financial economists to measure and describe the extent of the transmission of shocks between markets.

The GARCH (*p,q*) process proceeds as reproduced in Equations (1) and (2) [29].

$$\varepsilon_t \mid \psi_{t-1} \sim N(0, h_t) \tag{1}$$

$$h_t = \alpha_0 + \sum_{i=1}^{q} \alpha_i \varepsilon_{t-1}^2 + \sum_{i=1}^{p} \beta_i h_{t-1} \tag{2}$$

where $\varepsilon_t$ represents a stochastic process in discrete time, $\psi_{t-1}$ is information set through time $t$, and $h_t$ is the conditional volatility under the singularities of $p \geq 0$, $q > 0$, $\alpha_0 > 0$, $\alpha_1 \geq 0$, $i = 0, 1, \ldots, q$ and $\beta_i \geq 0$, $i = 1, 2, \ldots, p$. If $p = 0$ we have the ARCH (*q*) process in regression. The condition $\alpha + \beta < 1$ represents a stationary GARCH process in which the persistence in volatility is interpreted under the results of the coefficients $\alpha$ and $\beta$. In the situation where $\beta > \alpha$, it means that in the face of shocks with long-term effects, volatility does not decrease rapidly.

Later developments restricted the degree of interaction between the variables, allowing, in turn, emphasis of the covariances (or correlations) between the variables of interest. The conditional constant correlation (CCC) models and the dynamic conditional correlation (DCC) are introduced [33,34].

Using the DCC model allows for the estimation of variable coefficients in the form presented in Equation (3) [35].

$$\mathsf{N}_t = H_{xx,t}^{-1} h_{xy,t} \tag{3}$$

Being $H_{xx,t}^{-1}$ and $h_{xy,t}$ partitions of $H_t$ which represents an array $(H_{xx,t})$ with the structure of variances and covariances between the regressors and a vector $h_{xy,t}$ which contains, for each moment, the covariances of the dependent variable with the regressors included in the equation which defines the mean.

Another GARCH methodology that became known by the acronym BEKK (Baba, Engle, Kraft, and Kroner) was perfected by Engle and Kroner [36]. These last authors proposed a parameterization that easily restricts the requirements of $H_t$ to be positive for all values of $E_t$ and $x_t$ in the sample space. This model can be written in the following Equation (4).

$$H_t = CC' + \sum_{j=1}^{q} \sum_{k=1}^{K} A'_{kj} r'_{t-j} A_{jk} + \sum_{j=1}^{p} \sum_{k=1}^{K} B'_{kj} H_{i-j} B_{kj} \tag{4}$$

where $A_{jk}$, $B_{kj}$, and $C$ are N × N parameter matrices and $C$ is lower triangular. Finally, the decomposition of the constant term into a double matrix product aims to guarantee the positivity of $H_t$.

Multivariate GARCH (MGARCH) models are an extension of univariate GARCH models. The MGARCH models can be represented by a VARX (l,s) model as in Equation (5) [29].

$$y_t = \mu + C_1 y_{t-1} + \cdots + C_l y_{t-1} + D_0 x_t + \cdots + D_s x_{t-s} + \varepsilon_t \tag{5}$$

where $x_t$ corresponds to a vector of exogenous variables of dimension M and $C_i$ and $D_i$ represent coefficient matrices of dimensions N × N and N × M respectively.

Analyzing time dependence considering that financial volatilities move together over time between assets and markets, leads to more relevant empirical models than working with univariate analyses. The integration of financial markets is increasing and the participating agents (institutional and private investors) seek benefits through international diversification [37].

Since the emergence of multivariate analysis, the GARCH family models are increasingly used to describe and predict changes in the volatilities of time series, especially financial ones, gaining a new increase in use with the emergence of multivariate analysis [37].

In multivariate time series models, the impulse response function is used to analyze the effect of when a shock on a system series. This concept is associated with the modeling of the first moment of the series and can be generalized to the second moment using MGARCH models, and they define the volatility impulse response function, *VIRF*, for an MGARCH model, as the difference between the conditional and expected value expressed by the following expression exposed in Equation (6) [38].

$$VIRF_t\left(\delta\right) = E\left[vech\left(H_{t)}\right)\Big|\varepsilon_0 = \delta,\ \psi_{-1}\right] - E[vech\left(H_t\right)\Big|\varepsilon_0 = 0,\ \psi_{-1}] \tag{6}$$

where *VIRF* represents the effect that a shock of magnitude $\delta$ exhibits on the conditional variance, *t*, periods after it has occurred.

Several variants of GARCH models have emerged in the literature, but the most popular ones are the diagonal BEKK versions, the Constant Conditional Correlation (CCC), and the Dynamic Conditional Correlation (DCC) [39].

## 3. Data, Variables, Statistics, and Correlations

In this paper, we intend to verify the possibility of the performance of the stock indices of European countries that have companies that are part of the EURO STOXX Sustainability Index contaminating the volatility of this index. For this purpose, we used the valuations of the EURO STOXX Sustainability Index, for the period between 3 July 2000 and 28 February 2020 (i.e., a period greater than 20 years before the COVID-19 pandemic).

Additionally, to understand if there was any change in the contribution to the model after the COVID-19 pandemic declaration issued by the World Health Organization, we estimated a second model with all data between 3 July 2000 and 30 September 2022. The data were obtained from investing.com (accessed on 3 October 2022) and stoxx.com (accessed on 3 October 2022), which are free to access.

As mentioned above, the EURO STOXX Sustainability Index only includes 226 companies from 11 countries in the euro area which are listed on the stock exchange. These companies are chosen based on strict sustainability criteria at an economic, social, and environmental level. In comparison with other existing sustainability indices, the EURO STOXX Sustainability Index, for the attribution of scores in addition to integrally using sustainability, is concerned that companies create long-term value in the three aspects considered.

Table 1 contains the main statistics used in the entire sample for the original series. We can verify the high values of standard deviations, motivated by the various volatilities that occurred during the long period under analysis, which included the subprime crisis, sovereign debt, the COVID-19 pandemic, and now the war in Ukraine.

Table 1 also contains the results of the *p*-value of the Jarque–Bera test and the asymmetry and kurtosis statistics. We performed the Jarque–Bera test (in which all *p* values are less than 0.05, which implies the rejection of the normality hypothesis), to determine the normality of the series, together with kurtosis and asymmetry. As we can see, the variables do not follow a normal distribution, as the data are mostly skewed to the left. As the statistics are all less than three, our variables are platykurtic, that is, the value of excess kurtosis is negative, which means that the distributions have thinner tails than normal.

The countries where stock market indices have the highest standard deviations are Germany, Spain, Portugal, and Finland. In the opposite direction are the stock indices of Austria and Ireland. Regarding the EURO STOXX Sustainability Index, we found that it reached a maximum value of 160 points in November 2021 and a minimum value of 57 points in March 2009. Its standard deviation was 23.658 points.

To obtain accurate results from the empirical analysis, we also consider the problem of multicollinearity. The Pearson's correlation test (Table 2), applied to our variables, showed that there is no multicollinearity between the variables considered, considering that we used the value of 0.80 as a limit, similar to other studies [40]. We can also verify that we

only have two negative correlations, which are between the Portuguese and German, and Finnish stock market indices.

**Table 1.** Main descriptive statistics.

|  | Maximum | Minimum | Average | Std Deviation | Skewness | Kurtosis | Jarque–Bera |
|---|---|---|---|---|---|---|---|
| STOXX | 160,00 | 57.000 | 103.54 | 23.658 | 0.2531 | −0.7001 | 0.0000 |
| PRT | 13,688 | 3510.0 | 6580.2 | 2193.5 | 1.1735 | 0.7147 | 0.0000 |
| NLD | 829.10 | 194.30 | 423.12 | 125.19 | 0.5551 | −0.1395 | 0.0000 |
| ITA | 4339.6 | 966.91 | 2410.4 | 825.24 | 0.5214 | −0.5458 | 0.0000 |
| DEU | 16,290 | 2188.2 | 7599.3 | 3521.1 | 0.4899 | −0.8935 | 0.0000 |
| ESP | 16,040 | 5266.3 | 9740.1 | 3212.4 | 1.1731 | 0.9489 | 0.0000 |
| FRA | 7385.4 | 2401.5 | 4397.5 | 957.38 | 0.3157 | −0.6204 | 0.0000 |
| BEL | 4759.4 | 1425.9 | 3050.8 | 735.58 | −0.0301 | −0.9812 | 0.0000 |
| AUT | 648.95 | 138.35 | 301.25 | 109.18 | 1.1804 | 0.9687 | 0.0000 |
| IRL | 681.32 | 112.22 | 335.12 | 129.18 | −0.2358 | −0.6514 | 0.0000 |
| FIN | 5706.5 | 1085.1 | 2650.3 | 1127.7 | 0.5951 | −0.4674 | 0.0000 |
| LUX | 2590.0 | 654.00 | 1435.3 | 368.41 | 0.5925 | 0.7814 | 0.0000 |

Source: Authors' calculations. PRT–Portugal (PSI 20); NLD–The Netherlands (AEX); ITA–Italy (Italy 40); DEU–Germany (DAX); ESP–Spain (IBEX 35); FRA–France (CAC 40); BEL–Belgium (BEL 20); AUT–Austria (FTSE Austria); IRL–Ireland (FTSE Ireland); FIN–Finland (OMX Helsinki 25); LUX–Luxembourg (LUXXX).

**Table 2.** Pearson's correlation coefficients.

|  | STOXX | PRT | NLD | ITA | DEU | ESP | FRA | BEL | AUT | IRL | FIN | LUX |
|---|---|---|---|---|---|---|---|---|---|---|---|---|
| STOXX | 1 | 0.14 | 0.76 | 0.77 | 0.71 | 0.34 | 0.74 | 0.71 | 0.35 | 0.67 | 0.74 | 0.58 |
| PRT | - | 1 | 0.04 | 0.35 | −0.42 | 0.71 | 0.23 | 0.19 | 0.58 | 0.43 | −0.31 | 0.58 |
| NLD | - | - | 1 | 0.73 | 0.71 | 0.18 | 0.74 | 0.76 | 0.26 | 0.65 | 0.77 | 0.43 |
| ITA | - | - | - | 1 | 0.51 | 0.48 | 0.69 | 0.71 | 0.57 | 0.71 | 0.65 | 0.62 |
| DEU | - | - | - | - | 1 | 0.08 | 0.65 | 0.61 | 0.09 | 0.12 | 0.74 | 0.31 |
| ESP | - | - | - | - | - | 1 | 0.38 | 0.51 | 0.71 | 0.42 | 0.19 | 0.70 |
| FRA | - | - | - | - | - | - | 1 | 0.68 | 0.40 | 0.58 | 0.71 | 0.60 |
| BEL | - | - | - | - | - | - | - | 1 | 0.67 | 0.62 | 0.71 | 0.70 |
| AUT | - | - | - | - | - | - | - | - | 1 | 0.45 | 0.29 | 0.47 |
| IRL | - | - | - | - | - | - | - | - | - | 1 | 0.20 | 0.40 |
| FIN | - | - | - | - | - | - | - | - | - | - | 1 | 0.41 |
| LUX | - | - | - | - | - | - | - | - | - | - | - | 1 |

Source: Authors' calculations. PRT–Portugal; NLD–The Netherlands; ITA–Italy; DEU–Germany; ESP–Spain; FRA–France; BEL–Belgium; AUT–Austria; IRL–Ireland; FIN–Finland; LUX–Luxembourg.

## 4. GARCH Models Prerequisites

Before estimating the models of the GARCH family, it is first necessary to ensure the presence of three essential requirements: (i) Test the stationarity of the variables, in which the models developed by Dickey and Fuller [41] (the Augmented Dickey–Fuller–ADF and Kwiatkowski–Phillips–Schmidt–Shin (KPSS)) tests are the most used to determinate the lag order of the variables with stationarity [42]; (ii) The presence of persistent volatility clustering should be observed, i.e., big changes tend to be followed by big changes, whatever the sign and small changes tend to be followed by small changes [43]; (iii) There must be the presence of ARCH effects, which can be detected, for example, by the ARCHLM test. These three preconditions are usually called stylized facts for GARCH models.

### 4.1. Analysis of the Stationarity of Variables

The ADF test is based on the null hypothesis that the series is stationary against the alternative hypothesis that the series has a unit root. In case of $t_{stat} < t_{crit}$ we do not reject the null hypothesis, which means that the series is stationary. The KPSS test uses the Lagrange Multiplier (LM) statistic, and the test statistic is compared with the critical value for the desired significance level.

As we mentioned, we performed the Augmented Dickey–Fuller (ADF) and Kwiatkowski–Phillips–Schmidt–Shin (KPSS) tests, the results of which are shown in Table 3.

**Table 3.** Unit root tests.

| | ADF Test ($p$-Value) | | KPSS Test ($p$-Value) | |
|---|---|---|---|---|
| **Variables** | **Level** | **Ln First Diff.** | **Level** | **Ln First Diff.** |
| STOXX | 0.318 | <0.01 | <0.01 | >0.1 |
| PRT | 0.125 | <0.01 | <0.01 | >0.1 |
| NDL | 0.547 | <0.01 | <0.01 | >0.1 |
| ITA | 0.357 | <0.01 | <0.01 | >0.1 |
| DEU | 0.658 | <0.01 | <0.01 | >0.1 |
| ESP | 0.327 | <0.01 | <0.01 | >0.1 |
| FRA | 0.347 | <0.01 | <0.01 | >0.1 |
| BEL | 0.224 | <0.01 | <0.01 | >0.1 |
| AUT | 0.187 | <0.01 | <0.01 | >0.1 |
| IRL | 0.380 | <0.01 | <0.01 | >0.1 |
| FIN | 0.814 | <0.01 | <0.01 | >0.1 |

Source: Authors' calculation. Notes: <0.01 means lower than the 1% significance level; >0.1 means higher than 10% significance level. STOXX–EURO STOXX Sustainability Index; PRT–Portugal; NLD–The Netherlands; ITA–Italy; DEU–Germany; ESP–Spain; FRA–France; BEL–Belgium; AUT–Austria; IRL–Ireland; FIN–Finland; LUX–Luxembourg.

As can be seen in Table 3, by the results of the ADF test, the original variables do not show stationarity at the first level (recurrent situation in the financial time series) ($p$-value > 0.05), so it was necessary to resort to calculous of the natural logarithm of the first differences, to obtain stationarity of the variables. In other words, returns are used in the models. The KPSS test confirms the previous results, so, using the natural logarithm of the first differences, we do not fail to reject H0.

### 4.2. Testing the Presence of Persistent Volatility Clustering

Examining the following charts, we see the existence of persistent volatility on several occasions at the same time. The most significant volatilities occurred between 2008 and 2009 due to the subprime financial crisis and around March 2020, when the World Health Organization declared COVID-19 a pandemic [6]. There is also simultaneous volatility at the beginning of this century, although less intense, which was due to the terrorist attack on the Twin Towers in New York in September 2001.

Despite the graphical evidence (Figures 1 and 2) of the simultaneous existence of several volatilities, we performed the White test to see the possible existence of heteroscedasticity. The $p$-value of the test is 0.000, so we reject the null hypothesis of homoscedasticity, as the variance of the error term is not constant, which reinforces the justification of the GARCH modeling [28].

### 4.3. Verifying the Presence of ARCH Effects

Using the ARCH LM test, we can see in Table 4 that the results obtained show the existence of ARCH effects once the $p$-value of the test is <0.05. Therefore, we do not reject the null hypothesis that there are no ARCH effects.

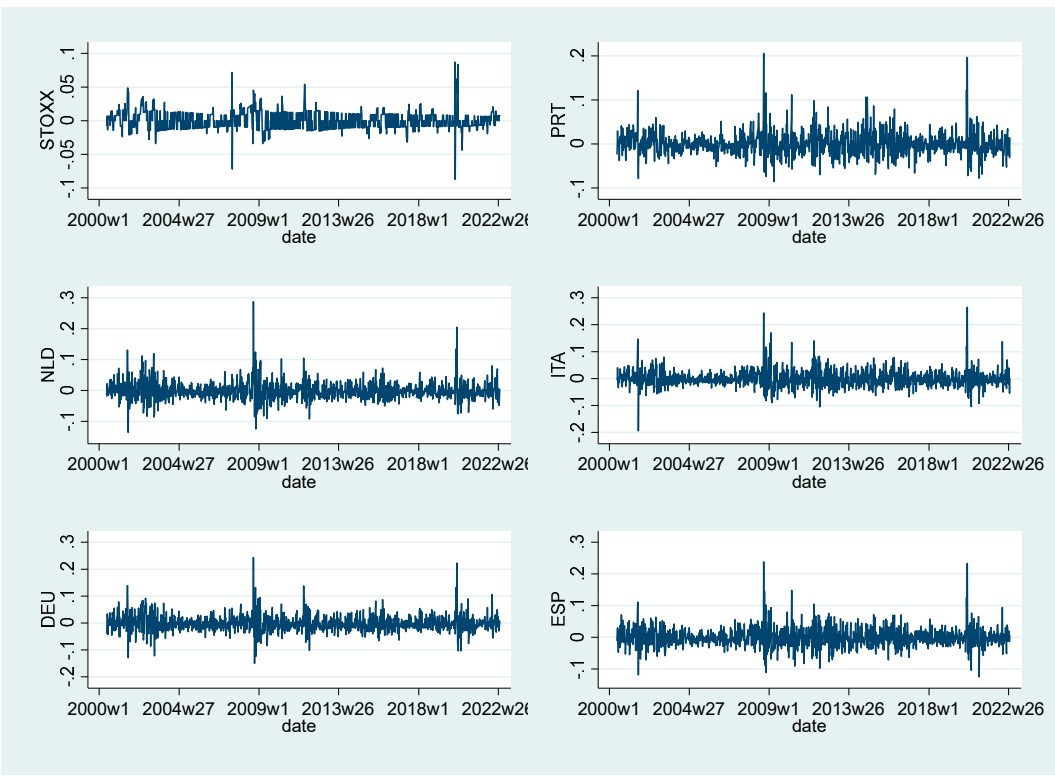

**Figure 1.** Volatility of variables STOXX, PRT, NLD, ITA, DEU, and ESP. Notes: Own elaboration. STOXX-EURO STOXX Sustainability Index; PRT-Portugal; NLD-The Netherlands; ITA-Italy; DEU-Germany; ESP-Spain.

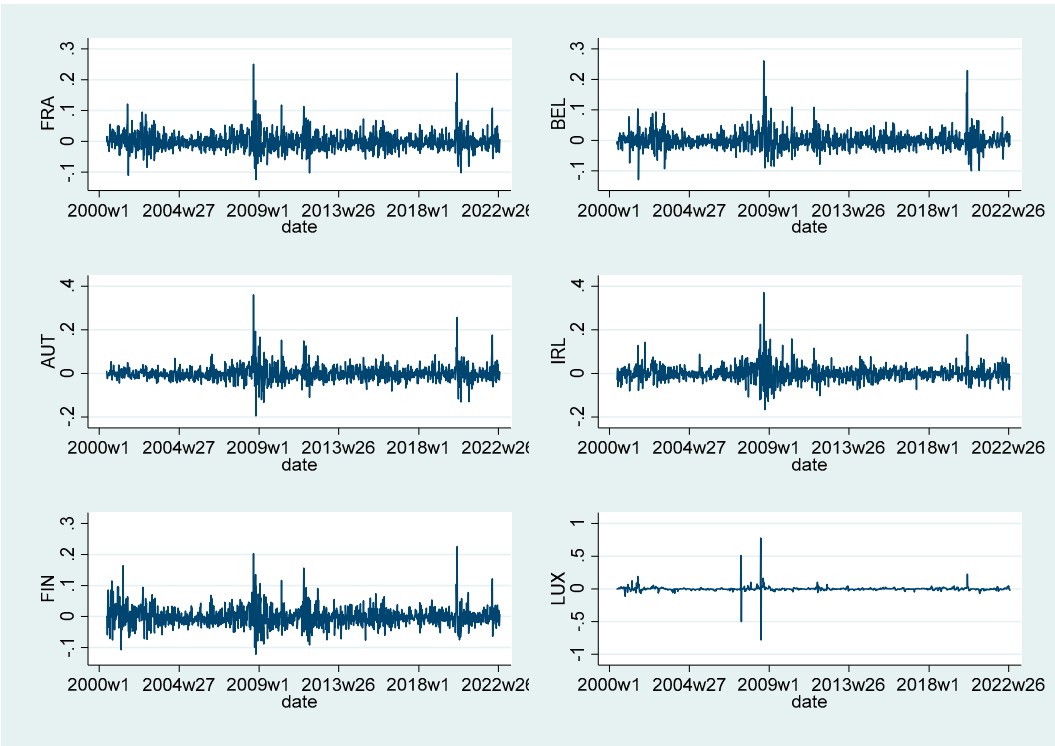

**Figure 2.** Volatility of variables FRA, BEL, AUT, IRL, FIN, and LUX. Notes: Own elaboration. FRA-France; BEL-Belgium; AUT-Austria; IRL-Ireland; FIN-Finland; LUX-Luxembourg.

**Table 4.** LM test for autoregressive conditional heteroskedasticity (ARCH).

| Lags (p) | Chi2 | df | Prob > Chi2 |
|----------|---------|----|-------------|
| 1 | 238.028 | 1 | 0.0000 |
| 2 | 251.123 | 2 | 0.0000 |
| 3 | 278.789 | 3 | 0.0000 |

Source: Authors' calculation.

Thus, the three prerequisites for the use of the GARCH family models are verified.

## 5. Baseline Models

As we mentioned earlier, the baseline models correspond to Multivariate GARCH models. Once we have verified the existence of ARCH effects at least until the third lag, we will estimate the MGARCH (1,1), MGARCH (1,2), and MGARCH (1,3) models, first for the pre-COVID-19 period and then for the entire sample, using the Dynamic Conditional Correlation (DCC).

Since we are studying a long time series, we intend to study the effects of the pandemic on the long-term model. The results obtained are shown in Table 5. Models 1, 2, and 3 comprise the estimates, respectively, for the MGARCH (1,1), MGARCH (1,2), and MGARCH (1,3) models, for the period between 3 July 2000 and 28 February 2020. In turn, models 4, 5, and 6 follow the same methodology, but for the entire sample.

**Table 5.** Estimation results.

| Variables | Model 1 | Model 2 | Model 3 | Model 4 | Model 5 | Model 6 |
|-----------|---------|---------|---------|---------|---------|---------|
| | MGARCH (1,1) | MGARCH (1,2) | MGARCH (1,3) | MGARCH (1,1) | MGARCH (1,2) | MGARCH (1,3) |
| STOXX | | | | | | |
| PRT | 0.0210 ** | 0.0201 ** | 0.0172 ** | 0.0147 *** | 0.0141 *** | 0.0101 *** |
| NLD | 0.0283 ** | 0.0308 *** | 0.0274 ** | 0.0254 ** | 0.0247 *** | 0.0258 *** |
| ITA | 0.0238 ** | 0.0479 ** | 0.0547 * | 0.0187 * | 0.0154 * | 0.0137 * |
| DEU | 0.0347 *** | 0.0310 ** | 0.0295 ** | 0.0387 ** | 0.0315 ** | 0.0374 *** |
| ESP | −0.0037 | −0.0021 | −0.0107 | −0.0023 | −0.0020 | −0.0017 |
| FRA | 0.0619 ** | 0.0547 * | 0.0508 * | 0.0635 * | 0.0574 * | 0.0599 * |
| BEL | 0.0057 | 0.0089 | 0.0147 | 0.0021 | 0.0019 | 0.0018 |
| AUT | 0.0414 ** | 0.0478 ** | 0.0405 * | 0.0408 ** | 0.0401 ** | 0.0409 ** |
| IRL | 0.0078 | 0.0061 | 0.0064 | 0.0935 | 0.0814 | 0.0714 |
| FIN | −0.0031 ** | −0.0039 ** | −0.0037 ** | −0.0047 * | −0.0038 * | −0.0074 * |
| LUX | 0.0701 *** | 0.0787 *** | 0.0615 ** | 0.0901 ** | 0.0847 ** | 0.0571 ** |
| ARC STOXX | | | | | | |
| ARCH L1 | 0.3354 *** | 0.3505 *** | 0.3471 *** | 0.3914 *** | 0.3714 *** | 0.3514 *** |
| GARCH L1 | 0.4714 *** | 0.2358 *** | 0.1834 *** | 0.4147 *** | 0.4011 *** | 0.3854 *** |
| GARCH L2 | | 0.1518 *** | 0.1594 *** | | 0.1381 *** | 0.1278 *** |
| GARCH L3 | | | 0.1254 *** | | | 0.1094 *** |
| Constant | 0.0008 * | 0.0009 ** | 0.0008 ** | 0.0009 ** | 0.0006 *** | 0.0009 *** |
| Observations | 5001 | 5001 | 5001 | 5697 | 5697 | 5697 |
| BIC | 6325.19 | 6158.94 | 6110.57 | 6251.14 | 6210.19 | 6138.12 |
| AIC | 6471.35 | 6378.15 | 6007.87 | 6314.10 | 6304.74 | 6247.85 |

Source: Authors' calculations. Notes: *, **, *** indicates 10%, 5% and 1%, respectively significance level. STOXX–EURO STOXX Sustainability Index; PRT–Portugal; NLD–The Netherlands; ITA–Italy; DEU–Germany; ESP–Spain; FRA–France; BEL–Belgium; AUT–Austria; IRL–Ireland; FIN–Finland; LUX–Luxembourg.

As can be seen in Table 5, the coefficients of the different models are similar and there are few variations in statistical significance. In all the models considered, there are only three stock indices that do not present statistical significance, which is the case of Spain, Belgium, and Ireland. We also verified that in all models, only the Finland index has a negative coefficient, which means that the relationship of its stock index with the Euro STOXX Sustainability Index is inverse.

To choose the best GARCH (*p,q*) model in each case, we perform the AIC test (Akaike's information criterium) and the BIC (Bayesian information criterium). The best choice will be the one in which the AIC and/or BIC criteria present the lowest results, because in this case, less information will be lost, and the quality of the models will be better. These statistical tests are widely used for economic and financial time series [44].

As we can see in Table 5, considering the AIC and BIC criteria, for the sample between 3 July 2000 and 28 February 2020, the best model is the GARCH (1,3). Following the same criteria, for the sample between 3 July 2000 and 30 September 2022, the best model is also the GARCH (1,3).

## 6. Discussion

Given the results obtained, we can visualize several important realities. In the first place, and generally, in all models, there is a high and positive significant effect of contagion between the stock indices of the eleven countries that make up the EURO STOXX Sustainability Index and the value of the same index. The independent variables clearly explain the volatility of the EURO STOXX Sustainability Index, that is, there are strong external influences of the variables on the volatility of the index. Partially, our results contradict those of Sharma et al. [10] and Jain et al. [25] considering that the authors conclude that there are no significant differences in the performance of sustainability indices and traditional ones. If they behaved commonly, no external differences would be revealed, which is not the case, as the current results indicate.

We can also verify that, contrary to most of the stock indices considered, in all the models, the stock indices of Spain, Ireland, and Belgium do not present statistical significance for the volatility of the EURO STOXX Sustainability Index. On the other hand, and surprisingly, the Finland stock index is in a counter-cycle with the volatility of the EURO STOXX Sustainability Index in all the estimated models. Therefore, risk-adjusted opportunities may be identified for investors while including sustainable investments in their portfolios, jointly with traditional market indexes, as also stated by Cunha et al. [9].

Of the stock market indices that positively explain the volatility of the EURO STOXX Sustainability Index, and given the GARCH model (1,3) for the period before the COVID-19 pandemic, the Luxembourg and Italian stocks index are the ones that most contribute to this volatility. In opposite terms, what contributes the least is the stock market index of Portugal. Regarding the total sample, the stock market index that most contributes to volatility is the French and the one that contributes the least is still the Portuguese. Important sustainability indexes studies regarding their volatile behavior exploration have been identified in the literature review, although with a different approach, not considering these volatility transmission effects, as explored presently [11,23,24,26,28].

In all estimated models, the volatility that occurs in the previous days in the EURO STOXX Sustainability Index affects the volatility of that same index in the following days, which may mean that internal contagions remain in the days immediately following the existence of that same volatility. Concerning the two-day lag, the internal influences keep statistical significance, to levels normally considered, and in the third lag, these influences keep to statistical significance, although with a lower intensity.

Considering only the model chosen through the AIC and BIC criteria, for the period from July 2000 until the declaration by the World Health Organization of the COVID-19 pandemic, the GARCH effect is verified in the following days. The volatility of the previous days in the EURO STOXX Sustainability Index, in the immediately following days, affects the value of the EURO STOXX Sustainability Index. For lags of two days, statistical significance remains concerning the contagion of this volatility.

Regarding the three-day lag, the volatility of the stock indices continues to show statistical significance and there is contagion in the EURO STOXX Sustainability Index, although with a lower intensity than the one-day lag. Despite the passage of time, the volatility of stock indices can still influence the evolution of the EURO STOXX Sustainability Index.

Concerning the total sample, that is, with the influences of the volatility of the stock market indices at the time of the COVID-19 pandemic, the evidence that is verified is practically like that found previously, although two changes are observed. The coefficients associated with the Dutch, Austrian, Portuguese, and German stock market indices reinforce their statistical significance. On the contrary, Finland's stock index reduces its statistical significance. As such, we are not able to confirm the results of Umar and Gubareva [12] stating that there is a diversification potential of ESG investments during pandemics due to the increased ESG indexes' volatility. Due to data restrictions, it was not possible to consider solely this period in the present study.

As in model 6, in this case, there are significant GARCH effects in the first, second, and third lags, although with lower intensities. We can also verify that perhaps because the volatility that we visualized in the graphs at the time of the pandemic was lower than the volatility seen in the 2007 and 2008 financial crises, the internal effects of the lags reduced their intensity, both for the first lag and for the second, despite maintaining their statistical importance.

The statistical significance presented by the EURO STOXX Sustainability Index lags is considered internal influences that influence the index itself, but which originated from the contagion of influences external to the index, which are the stock market indices. In the case of France, Germany, and Austria, in the estimation of the total sample, the intensities of influence of the stock indices on the EURO STOXX Sustainability Index were reinforced. In the case of Portugal, Holland, Italy, and Luxembourg, these intensities were reduced. Despite this result, we can argue that ESG investments contribute to financial stability and performance [2], and their availability emergence was justified [4], offering good diversification potential and valuable risk-adjusted opportunities to investors' portfolios.

## 7. Conclusions

This paper uses MGARCH modeling to study the possible presence of volatility contagion effects between the stock indices of 11 countries and the EURO STOXX Sustainability Index.

One of the concerns that one must have when defining sustainability is not to let there be an excessive influence of economic and financial aspects on that same sustainability.

Despite the EURO STOXX Sustainability Index being made up of the traditional three aspects of sustainability: economic, social, and environmental (triple bottom line), the contagion of the financial markets in fixing the value of the index is strong and significant, which may mean that the social components and environmental aspects of the index are being undervalued, with only one component of the triple bottom line being given high importance. Perhaps here we can speak of an exaggerated financial influence of the stock markets in the EURO STOXX Sustainability Index, leaving social and environmental sustainability in the background. We cannot forget that, mainly during the financial crises of 2007 and 2008 and during the COVID-19 pandemic, these European countries revealed enormous social concerns to offset the pernicious effects of these crises on their populations.

The internal elements of volatility that occur in the EURO STOXX Sustainability Index, are felt intensely and significantly in the days immediately following the occurrence of shocks in the stock indices. They remain statistically significant on subsequent days, but the intensity is reduced. Perhaps here we can speak of a memory effect that tends to diminish over time.

In addition to the strong and incomprehensible contagion of stock indices in the EURO STOXX Sustainability Index, there is also a significant influence of the index's internal effects on the index itself, especially in the first and second lags.

This study also makes it possible to open up the discussion on the influence of stock markets on sustainability indices, and the results can make it possible to bring new contributions to the formulation of these indices, which should not be excessively contaminated by the equity markets, otherwise, the concept of sustainability would only be focused on economic and financial aspects.

One of the suggestions for future work may be to verify whether this evidence occurs in the same or similar way in other sustainability indices, or even in merely financial or economic indices, broadening the empirical debate on the subject. Policymakers, portfolio managers, and investors should consider these effects while composing their portfolios, and greenwashing possibilities' inclusion would be a valuable contribution to the emergent literature on the topic.

**Author Contributions:** Conceptualization, M.C.N. and M.M.; investigation, M.C.N. and M.M.; methodology, M.C.N.; supervision, M.M.; writing—original draft, M.C.N.; writing—review and editing, M.M. All authors have read and agreed to the published version of the manuscript.

**Funding:** This research received no external funding.

**Institutional Review Board Statement:** Not applicable.

**Informed Consent Statement:** Not applicable.

**Data Availability Statement:** Not applicable.

**Conflicts of Interest:** The authors declare no conflict of interest.

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
