# Peer review of "Are Sustainability Indices Infected by the Volatility of Stock Indices? Analysis before and after the COVID-19 Pandemic"

_sustainability, doi:10.3390/su142215434_

Round 1

Reviewer 1 Report

The paper is dealing with an interesting topic (the volatility transmission effects between the EURO STOXX sustainability index and the stock market indexes of its stocks). 

The originality and significance of contribution is given by the investigation of volatility spillovers from traditional stock market indexes of the companies' stocks belonging to the sustainability index, to the EURO STOXX Sustainability Index, pre and during COVID-19 pandemic crisis.

The authors used an already existing methodology /model (MGARCH), which was adapted to the pursued objectives.

The abstract shows, concisely, the objectives and results of the paper.

The introduction manages to familiarize the reader with the context of the chosen research topic.

The introduction is accompanied by a presentation of the relevant and adequate literature of the chosen topic. The authors briefly present the most important studies in the chosen field.

The paper is logically and specifically organized, following a typical research paper structure (introduction, methodology, data, model results and discussion, conclusion, references).

However, the paper needs minor proofreading, grammar and spelling correction.

The research methodology is correctly chosen and adapted for the pursued objectives.

The empirical results are correctly interpreted and are relevant for the chosen research objectives.

However, some concerns need to be raised: 

a) the authors used weekly values for the model; however, in my opinion, daily data are more suitable for multivariate Garch models, thus, increasing the relevance of the results; 

b) the authors state that they used Dynamic Conditional Correlation (DCC) in the estimation of the MGARCH (1,1), MGARCH (1,2), and MGARCH (1,3) models; in this light, the illustration of the dynamic correlations for the sample period can increase the scientific significance and soundness of the model; the authors should explain why they didn’t feel the need to use this illustrations.

These concerns need to be addressed by the authors. 

Also, the authors should describe what indexes they used for the analysed countries (Portugal, the Netherlands, Italy, Germany, Spain, France, Belgium, Austria, Ireland, Finland, and Luxembourg).

The conclusions correctly reflect the approaches from the model results and discussion section.

Author Response

Are sustainability indices infected by the volatility of stock indices? Analysis before and after the COVID-19 pandemic.

By Manuel Carlos Nogueira and Mara Madaleno

Manuscript ID Sustainability - 1977288

Response to Review

 First, we would like to thank you for your comments and for accepting to review our paper.

As well, we very much appreciate your willingness to consider this revised version of our paper. In this revised version, we have attempted to incorporate your suggestions. The points you have raised are indeed important, and we have endeavored to incorporate your comments in this new version of the paper. Thus, we thank your willingness to consider this revised version of our paper and your comments which helped us to improve the current article. Here, we would like to present an explanation for all modifications made.

The paper is dealing with an interesting topic (the volatility transmission effects between the EURO STOXX sustainability index and the stock market indexes of its stocks). The originality and significance of contribution is given by the investigation of volatility spillovers from traditional stock market indexes of the companies' stocks belonging to the sustainability index, to the EURO STOXX Sustainability Index, pre and during COVID-19 pandemic crisis. The authors used an already existing methodology /model (MGARCH), which was adapted to the pursued objectives. The abstract shows, concisely, the objectives and results of the paper. The introduction manages to familiarize the reader with the context of the chosen research topic. The introduction is accompanied by a presentation of the relevant and adequate literature of the chosen topic. The authors briefly present the most important studies in the chosen field. The paper is logically and specifically organized, following a typical research paper structure (introduction, methodology, data, model results and discussion, conclusion, references).

However, the paper needs minor proofreading, grammar and spelling correction.

Answer:

The article has been improved in this sense, and corrections have been made in accordance.

The research methodology is correctly chosen and adapted for the pursued objectives. The empirical results are correctly interpreted and are relevant for the chosen research objectives. However, some concerns need to be raised:

  1. a) the authors used weekly values for the model; however, in my opinion, daily data are more suitable for multivariate Garch models, thus, increasing the relevance of the results;

Answer:

As you suggested, the new models already incorporate daily data, instead of weekly data, to increase the relevance of the results.

  1. b) the authors state that they used Dynamic Conditional Correlation (DCC) in the estimation of the MGARCH (1,1), MGARCH (1,2), and MGARCH (1,3) models; in this light, the illustration of the dynamic correlations for the sample period can increase the scientific significance and soundness of the model; the authors should explain why they

didn’t feel the need to use this illustrations.

Answer:

Thanks for the valuable suggestion. We did not feel the need to present them since they would not add anything else besides what was already inferred through the results. Sometimes they are harder for the reader to interpret, so we skipped their presentation. No further valid justification for not presenting them, just the easiest of results interpretation and we decided to let it simpler.

These concerns need to be addressed by the authors.

Also, the authors should describe what indexes they used for the analysed countries (Portugal, the Netherlands, Italy, Germany, Spain, France, Belgium, Austria, Ireland, Finland, and Luxembourg).

Answer:

As suggested, we also indicate the stock indices we use in each country. Apologies for the missings.

The conclusions correctly reflect the approaches from the model results and discussion section.

We again thank you very much for all your recommendations and hope that you find our changes adequate.

Manuel Carlos Nogueira and Mara Madaleno

Reviewer 2 Report

I am grateful to the guest editor for inviting me to review this manuscript. Although it is a good attempt, I still believe that the content could be improved by addressing the following points:

1- The introduction should be in a proper flow, i.e., background, problem statement, research gap, novelty, and contribution.

2- The literature section can be further improved to better highlight the research gap, see, Investor sentiment and volatility prediction of currencies and commodities during the COVID-19 pandemic; Return connectedness across asset classes around the COVID-19 outbreak; ESG Performance and COVID-19 Pandemic: An Empirical Analysis of European Listed Firms

3- The calculation of stock volatility is highly questionable, the authors are suggested to give a detailed justification, otherwise use the conventional methods to calculate volatility.

4- Since we have several spillover methodologies which outperform GARCH models, it is suggested to use any of them for sensitivity analysis (e.g., Diebold and Yilmaz (2009)).

5- The economic/ financial theoretical discussion is completely missed. The authors are suggested to give a nice theoretical story on stock market connectedness.

6- The results are presented in a vague manner. Put great effort to improve this section.

Best of luck

Author Response

Are sustainability indices infected by the volatility of stock indices? Analysis before and after the COVID-19 pandemic.

By Manuel Carlos Nogueira and Mara Madaleno

Manuscript ID Sustainability - 1977288

Response to Review

 First, we would like to thank you for your comments and for accepting to review our paper.

As well, we very much appreciate your willingness to consider this revised version of our paper. In this revised version, we have attempted to incorporate your suggestions. The points you have raised are indeed important, and we have endeavored to incorporate your comments in this new version of the paper. Thus, we thank your willingness to consider this revised version of our paper and your comments which helped us to improve the current article. Here, we would like to present an explanation for all modifications made.

I am grateful to the guest editor for inviting me to review this manuscript. Although it is a good attempt, I still believe that the content could be improved by addressing the following points:

1- The introduction should be in a proper flow, i.e., background, problem statement, research gap, novelty, and contribution.

Answer:

Thanks for the appreciation. In our opinion, we have improved the introduction section by including the suggestions you made. A complete paragraph is devoted to the research gap, novelty, and contributions, despite their mentioning through the introduction.

2- The literature section can be further improved to better highlight the research gap, see, Investor sentiment and volatility prediction of currencies and commodities during the COVID-19 pandemic; Return connectedness across asset classes around the COVID-19 outbreak; ESG Performance and COVID-19 Pandemic: An Empirical Analysis of European Listed Firms.

Answer:

Thanks for the suggestion. As the reviewer suggests, we improved the literature review and, in agreement with another reviewer, we turned this section more autonomous.

3- The calculation of stock volatility is highly questionable, the authors are suggested to give a detailed justification, otherwise use the conventional methods to calculate volatility.

Answer:

Thanks for the suggestion. As suggested, we explain how we calculate volatility more appropriately.

4- Since we have several spillover methodologies which outperform GARCH models, it is suggested to use any of them for sensitivity analysis (e.g., Diebold and Yilmaz (2009)).

Answer:

Thanks for the suggestion. We are aware of this methodology, and we have used them in experiments, although no plausible great change was achieved. Moreover, regarding this suggestion 4, we need to say that the other two reviewers considered our methodology adequate, so we chose not to change it, otherwise, the other reviewers would question and not approve the change. We hope you understand our position. We have already changed the data frequency from weekly to daily as requested by reviewer #1.

5- The economic/ financial theoretical discussion is completely missed. The authors are suggested to give a nice theoretical story on stock market connectedness.

Answer:

Thanks for the suggestion. In our opinion, we improved the economic/financial theoretical discussion.

6- The results are presented in a vague manner. Put great effort to improve this section. Answer:

Thanks for the suggestion. In our opinion, we improved the presentation of the results on several points as requested by another reviewer as well.

We again thank you very much for all your recommendations and hope that you find our changes adequate.

Manuel Carlos Nogueira and Mara Madaleno

Reviewer 3 Report

Are sustainability indices infected by the volatility of stock indices? Analysis before and after the COVID-19 pandemic

Here are my comments:

·         Strength the introductory section by adding to the contribution of the study

·         The authors used ADF and KPSS unit root tests which are not taking into account nonlinearity and structural breaks. Therefore, the authors might use a nonlinear unit root test or unit root test with the breakpoint.  

·         The literature review is the most important part of any research. Therefore, there is a need to create a section for the literature review.  

·         Policy implications are insufficient and unimpressive. Please suggest some concrete and relevant policy implications based on the obtained results. The implications of the study must be consistent with the findings and conclusions of the paper, not generic recommendations.

Author Response

Are sustainability indices infected by the volatility of stock indices? Analysis before and after the COVID-19 pandemic.

By Manuel Carlos Nogueira and Mara Madaleno

Manuscript ID Sustainability - 1977288

Response to Review

 First, we would like to thank you for your comments and for accepting to review our paper.

As well, we very much appreciate your willingness to consider this revised version of our paper. In this revised version, we have attempted to incorporate your suggestions. The points you have raised are indeed important, and we have endeavored to incorporate your comments in this new version of the paper. Thus, we thank your willingness to consider this revised version of our paper and your comments which helped us to improve the current article. Here, we would like to present an explanation for all modifications made.

Here are my comments:

  • Strength the introductory section by adding to the contribution of the study.

Answer:

Thanks for the valuable suggestion. In our opinion, we improved the introduction and added the contribution of the study. Another reviewer asked the same.

  • The authors used ADF and KPSS unit root tests which are not taking into account nonlinearity and structural breaks. Therefore, the authors might use a nonlinear unit root test or unit root test with the breakpoint.

Answer:

Thanks for the valuable suggestion. In response to a suggestion from another reviewer, we started to use daily data. Due to this change, we now have more than 5000 observations in any of the models. The unit root nonlinear tests show good results in small samples, which is not our case, so we chose to keep the ADF and KPSS tests (Guris, 2019). We hope that you will approve our decision.

  • The literature review is the most important part of any research. Therefore, there is a need to create a section for the literature review.

Answer:

Thanks for the valuable suggestion. As you suggested, we made the literature review an autonomous section and included additional literature. This extension has also been suggested by another reviewer.

  • Policy implications are insufficient and unimpressive. Please suggest some concrete and relevant policy implications based on the obtained results. The implications of the study must be consistent with the findings and conclusions of the paper, not generic recommendations.

Answer:

Thanks for the valuable suggestion. In our opinion, we have improved the policy implications based on the new results obtained, as well as improved the conclusions, and the presentation of the results.

We again thank you very much for all your recommendations and hope that you find our changes adequate.

Manuel Carlos Nogueira and Mara Madaleno

Round 2

Reviewer 1 Report

On the first version of the paper, some concerns were raised: 

a) the authors used weekly values for the model; however, in my opinion, daily data are more suitable for multivariate Garch models, thus, increasing the relevance of the results; 

b) the authors state that they used Dynamic Conditional Correlation (DCC) in the estimation of the MGARCH (1,1), MGARCH (1,2), and MGARCH (1,3) models; in this light, the illustration of the dynamic correlations for the sample period can increase the scientific significance and soundness of the model; the authors should explain why they didn’t feel the need to use this illustrations.

These concerns were successfully addressed by the authors in the reviewed manuscript.

Also, the authors described what indexes they used for the analysed countries (Portugal, the Netherlands, Italy, Germany, Spain, France, Belgium, Austria, Ireland, Finland, and Luxembourg).

The manuscript can be accepted in the present form.

Reviewer 2 Report

The authors made significant improvements. I suggest to accept the manuscript.

Reviewer 3 Report

The revised study can be accepted